The affective profiles, psychological well-being, and harmony: environmental mastery and self-acceptance predict the sense of a harmonious life

Garcia Danilo 1 2 danilo.garcia@euromail.se
Al Nima Ali 1 3
Kjell Oscar N.E. 1 4
1 Network for Empowerment and Well-Being, University of Gothenburg , Gothenburg , Sweden
2 Center for Ethics, Law and Mental Health (CELAM), University of Gothenburg , Gothenburg , Sweden
3 Department of Psychology, University of Gothenburg , Gothenburg , Sweden
4 Department of Psychology, Lund University , Lund , Sweden
Abdullah Jafri
Electronic publication date: 2014 Feb 13
Publication date: 2014
Volume: 2
Electronic Location ID: e259
Received 2013 Dec 18; Accepted 2014 Jan 13
Copyright: © 2014 Garcia et al.
Copyright year: 2014
Copyright holder: Garcia et al.
License: This is an open access article distributed under the terms of the Creative Commons Attribution License, which permits unrestricted use, distribution, and reproduction in any medium, provided the original author and source are credited.
License URL: https://creativecommons.org/licenses/by/3.0/

Keywords: Affective profiles model, Positive and negative emotions, Well-being, Harmony in life, Environmental mastery, Psychological well-being, Self-acceptance, Environmental mastery

Funding: This study was supported by a grant to Danilo Garcia from AFA Insurance. The funders had no role in study design, data collection and analysis, decision to publish, or preparation of the manuscript.

==============================
Background. An important outcome from the debate on whether wellness equals happiness, is the need of research focusing on how psychological well-being might influence humans’ ability to adapt to the changing environment and live in harmony. To get a detailed picture of the influence of positive and negative affect, the current study employed the affective profiles model in which individuals are categorised into groups based on either high positive and low negative affect (self-fulfilling); high positive and high negative affect (high affective); low positive and low negative affect (low affective); and high negative and low positive affect (self-destructive). The aims were to (1) investigate differences between affective profiles in psychological well-being and harmony and (2) how psychological well-being and its dimensions relate to harmony within the four affective profiles.

Method. 500 participants (mean age = 34.14 years, SD. = ±12.75 years; 187 males and 313 females) were recruited online and required to answer three self-report measures: The Positive Affect and Negative Affect Schedule; The Scales of Psychological Well-Being (short version) and The Harmony in Life Scale. We conducted a Multivariate Analysis of Variance where the affective profiles and gender were the independent factors and psychological well-being composite score, its six dimensions as well as the harmony in life score were the dependent factors. In addition, we conducted four multi-group (i.e., the four affective profiles) moderation analyses with the psychological well-being dimensions as predictors and harmony in life as the dependent variables.

Results. Individuals categorised as self-fulfilling, as compared to the other profiles, tended to score higher on the psychological well-being dimensions: positive relations, environmental mastery, self-acceptance, autonomy, personal growth, and purpose in life. In addition, 47% to 66% of the variance of the harmony in life was explained by the dimensions of psychological well-being within the four affective profiles. Specifically, harmony in life was significantly predicted by environmental mastery and self-acceptance across all affective profiles. However, for the low affective group high purpose in life predicted low levels of harmony in life.

Conclusions. The results demonstrated that affective profiles systematically relate to psychological well-being and harmony in life. Notably, individuals categorised as self-fulfilling tended to report higher levels of both psychological well-being and harmony in life when compared with the other profiles. Meanwhile individuals in the self-destructive group reported the lowest levels of psychological well-being and harmony when compared with the three other profiles. It is proposed that self-acceptance and environmental acceptance might enable individuals to go from self-destructive to a self-fulfilling state that also involves harmony in life.

The affective profiles model is based on individuals’ affective experience and consist of four different profiles: self-fulfilling (high positive affect, low negative affect), high affective (high positive affect, high negative affect), low affective (low positive affect, low negative affect), and self-destructive (low positive affect, high negative affect) (see among others Norlander, Bood & Archer, 2002; Bood, Archer & Norlander, 2004; Norlander, Johansson & Bood, 2005; Archer et al., 2007; Karlsson & Archer, 2007; Palomo et al., 2007; Palomo et al., 2008; Archer, Adolfsson & Karlsson, 2008; Schütz, Garcia & Archer, 2014). The model discerns differences between profiles in measures of negative (i.e., ill-being) and positive (i.e., well-being) mental health (e.g., Garcia, 2011; Garcia, 2012; Garcia & Archer, 2012; Garcia et al., 2012; Garcia et al., 2010; Garcia & Siddiqui, 2009a; Garcia & Siddiqui, 2009b; Nima et al., 2013; Jimmefors, Garcia & Archer, in press). Importantly, this approach provides a more informative and detailed picture of the nature of positive and negative affect as compared with simply treating them as two separate variables or adding them together to one mean value (Garcia, 2011).

To the best of our knowledge, the affective profiles model has mostly been examined among Swedes. Nevertheless, the few studies employing the model in other populations show similar results (for studies using Indonesian, Iranian, Dutch, respectively US-residents see: Adrianson et al., 2013; Garcia & Moradi, 2013; Kunst, 2011; Schutz et al., 2013). In general, self-fulfilling individuals report feeling more energetic and optimistic than the other three affective profiles, while all four profiles react differently to stress and have different exercise habits and blood pressure (for a review see Garcia, Ghiabi et al., 2013). Self-fulfilling and high affective individuals show the best performance during stress, have a more active life, and lower blood pressure than individuals with low affective and self-destructive profiles (Norlander, Bood & Archer, 2002; Norlander, Johansson & Bood, 2005). Moreover, when compared to self-fulfilling and high affective individuals, low affective individuals have responded maladaptively to induced stress (Norlander, Bood & Archer, 2002); but at the same time low affectives report less stress in their life compared to high affective and low destructive individuals (Norlander, Johansson & Bood, 2005). Some researchers have suggested that low affective individuals “go their own way” when choosing their environment. In other words, low affectives are determined and autonomous when avoiding stressful situations in order to avoid pain and displeasure, but also to feel pleasure and satisfaction with their life (Garcia et al., 2010). However, low affective individuals seem to avoid positive meaningful experiences to maintain the status quo of their affectivity levels (i.e., low positive affect and low negative affect), which might give them a sense of balance in life (Garcia et al., 2010).

There is, however, a lack of studies in adult populations using the affective profiles model and positive measures of mental health. This is important because the absence of life satisfaction and positive emotions, for example, is more predictive of subsequent mortality and morbidity than the presence of negative emotions (Cloninger, 2004; Cloninger, 2006; Cloninger, 2013; Huppert & Whittington, 2003). In a recent study, Schutz and colleagues (Schutz et al., 2013) fill this gap in the literature by using a relatively large population of 1,400 US-resident who reported happiness, life satisfaction, and happiness-increasing strategies. Among US-residents, the self-fulfilling individuals reported significantly higher levels of happiness and significantly lower levels of depression than all the individuals in the other three groups (i.e., high affective, low affective, self-destructive). At the other end, the self-destructive group reported significantly higher levels of depression and lower levels of happiness than the other groups (i.e., self-fulfilling, high affective, low affective). These researchers concluded that positive affect might serve as an anti-depressive factor as well as a facilitative factor for happiness and life satisfaction (see also Archer & Kostrzewa, 2013; Archer et al., 2013; Lindahl & Archer, 2013).

In regards to happiness-increasing strategies, self-fulfilling individuals scored higher in strategies related to agentic (i.e., self-directedness: work on self-control, reach one’s full potential, organizing one’s life and goals, striving for accomplishment of tasks, proneness to wellness through fitness and flow), communal (i.e., cooperation: supporting and encouraging friends, helping others, interacting with friends, and receiving help from friends), and spiritual values (i.e., self-transcendence: seeking support from faith, performing religious activities, praying) (Schutz et al., 2013). These results are in line with findings about agency and communion’s association to mental health, dysfunction and suffering (Cloninger & Zohar, 2011; Garcia, 2012; Garcia, Anckarsaẗer & Lundstrom̈, 2013; Garcia, Lundström et al., 2013; Garcia, Nima & Archer, 2013) and their role in enabling individuals to become happier, healthier, and less depressed (Cloninger, 2013; Johansson et al., 2013). Schutz and colleagues (Schutz et al., 2013) suggest that differences between affective profiles imply that promoting positive emotions can positively influence a depressive-to-happy state as well as increasing life satisfaction. Moreover, these researchers suggest that the pursuit of happiness through agentic, communal, and spiritual values leads to a self-fulfilling experience defined as frequently experiencing positive emotions and infrequently experiencing negative emotions (see also Cloninger, 2013; Nima, Archer & Garcia, 2012; Nima et al., 2013).

In this article we address other positive measures of mental health, namely, psychological well-being and harmony in life. Although these measures are related to happiness (i.e., life satisfaction, positive and negative affect; Diener, 1984) they represent distinct conceptualisations of well-being, and thus, measured with different instruments. Psychological well-being, for instance, has been suggested as conceptually different from happiness because it defines intra-personal attributes related to adaptation, self-actualization, and empowerment (Garcia, 2011). An important outcome from the debate on whether wellness equals happiness (see Biswas-Diener, Kashdan & King, 2009; Delle Fave & Bassi, 2009; Kashdan, Biswas-Diener & King, 2008; Garcia, 2013; Ryan & Huta, 2009; Straume & Vittersø, 2012; Waterman, 2008), is the need of research focusing on how psychological well-being might influence humans’ ability to adapt to the changing environment and live in harmony. Next, we briefly review these two positive measures of mental health.

Psychological well-being

Ryff (1989) developed a multidimensional model of well-being called psychological well-being, which includes 6 dimensions: positive relations with others, environmental mastery, self-acceptance, autonomy, personal growth, and purpose in life (see Table 1 for definitions). These six dimensions define Ryff’s conceptualization of psychological well-being both theoretically and operationally, and they identify what promotes effective mastery of life and emotional and physical health (Ryff, 1989, Ryff, 1995). For example, among Swedish adolescents, psychological well-being, and especially the self-acceptance and environmental mastery dimensions, strongly relate to high levels of positive affect and life satisfaction (Garcia, 2011, 2012; Garcia & Archer, 2012; Garcia & Siddiqui, 2009b).

Table 1 Definition of the six dimensions of psychological well-being.

Psychological well-being dimension	Definition	
Self-acceptance	Emphasis on acceptance of the self and of one’s past life.	
Positive relations with others	Having strong feelings of empathy and affection for all human beings and as being capable of greater love, deeper friendship, and more complete identification with others and warm relating to others.	
Autonomy	Expressions of internal locus of evaluation, thus not looking to others for approval but evaluating oneself by personal standards.	
Environmental mastery	The individual’s ability to choose or create environments suitable to his or her psychic conditions.	
Purpose in life	Having goals, intentions, and a sense of direction, all of which contribute to the feeling that life is meaningful.	
Personal growth	Emphasis to continued growth and the confronting of new challenges or tasks at different periods of life.	

By employing the affective profiles model, researchers have found that self-fulfilling adolescents report higher levels on several of the psychological well-being dimensions. For example, Garcia & Siddiqui (2009b) found that environmental mastery was higher among self-fulfilling individuals as compared to all other profiles (see also Kjell et al., 2013b). An important observation is also that high and low affective groups differed from each other in psychological well-being dimensions associated to agentic values (e.g., high affectives reported higher personal growth than low affectives) not to those dimensions associated to communal values (i.e., positive relations with others). Purpose in life and personal growth are, indeed, distinctive to the other psychological well-being dimensions (Ryff & Singer, 1998; Ryff & Keyes, 1995)— that is, the pursuit of one’s true potential or one’s great life questions may at times not bring positive emotions and might distort the balance or status quo in one’s life. In most of the dimensions, however, the high and low affective individuals showed higher levels than the self-destructive.

Harmony in life

Harmony in life has been suggested as a complement to satisfaction with life (Kjell et al., 2013a). When measuring life satisfaction, individuals are asked to evaluate if their life is according to their expectations or an ideal (Diener, 1984; Diener et al., 1985). In this context, life satisfaction is seen as the cognitive part of happiness, while affect (i.e., positive and negative affect) is seen as the affective part. It has been argued that this evaluation does not by itself represent the full breadth of individuals’ cognitive well-being (Kjell, 2011). The assessment of harmony, in contrast, encourages individuals to assess their global, subjective perception of harmony in life; which includes a global and overall assessment of whether one’s life involve balance, mindful non-judgemental acceptance, fitting in and being attuned with one’s life. When comparing the two concepts using quantitative semantics on words that participants have generated to each term, reveals that the concept of satisfaction is significantly more related to achievement, education, work, money and car; whilst the concept of harmony is significantly more related to balance, peace, cooperation, agreement and meditation (Kjell et al., 2013a). Harmony and life satisfaction, as most well-being constructs, correlate with each other; but they are also distinct, the sense of a harmonious life explains unique variance in stress and depression (Kjell et al., 2013a). Furthermore, harmony, compared to life satisfaction, is more strongly related to the psychological well-being dimensions; meanwhile life satisfaction relates more strongly to happiness (Kjell et al., 2013b).

Although we have detailed a difference between harmony and life satisfaction, we expect similar results using harmony in life as a construct of cognitive well-being. In other words, individuals with a self-fulfilling profile are hypothesised to report higher levels of harmony in life than the other profiles. Further, as harmony in life and psychological well-being have been found to be particularly related, it is important to further investigate this. In particular, we expected harmony to be related with self-acceptance and environmental mastery among profiles. Although, low affective individuals might “go their own way” (i.e., involving high levels of autonomy or agentic values) when approaching pleasantness; their tendency to avoid pain and meaningful experiences (Garcia et al., 2010) is expected to lower other agentic dimensions of psychological well-being: personal growth and purpose in life. This in turn is expected to relate to a lower sense of a harmony in life.

The present study

The aims were:

1. To investigate differences between affective profiles in the different dimensions of psychological well-being and harmony in life.

2. To investigate how dimensions of psychological well-being relate to harmony in life within the affective profiles.

Method

Ethics statement

This research protocol was approved by the Ethics Committee of the University of Gothenburg. Participants consented to take part in the study.

Participants and procedure

The participants (N = 500, age mean  = 34.14 years sd.  = ±12.75 years; 187 males and 313 females) were recruited through Amazon’s Mechanical Turk (MTurk; https://www.mturk.com/mturk/welcome). MTurk allows data collectors to recruit participants (workers) online for completing different tasks for money (for a review on the validity of this method for data collection see Buhrmester, Kwang & Gosling, 2011). As in Schutz and colleagues’ (Schutz et al., 2013) study, participants in the present study were recruited by the following criteria: US-residency and fluency in English. Participants were paid a wage of two American dollars for completing the task and informed that the study was confidential and voluntary. The participants were presented with a battery of self-reports comprising the well-being measures, as well as questions pertaining age and gender in the following order: demographics, affect measure, psychological well-being scale, and harmony scale.

Instruments

The Positive Affect and Negative Affect Schedule (Watson, Clark & Tellegen, 1988). Participants are instructed to rate to what extent they generally have experienced 20 (10 positive and 10 negative) different feelings or emotions during the last weeks, using a 5-point Likert scale (1 = very slightly, 5 = extremely). The 10-item positive affect scale includes adjectives such as strong, proud, and interested (Cronbach’s α = .90). The 10-item negative affect scale includes adjectives such as afraid, ashamed and nervous (Cronbach’s α = .88).

The Scales of Psychological Well-Being (short version; Clarke et al., 2001). The instrument comprises 18 items using a 6-point Likert scale (1 = strongly disagree, 6 = strongly agree), 3 items for each of the 6 psychological well-being dimensions: (1) positive relations with others (e.g., “People would describe me as a giving person, willing to share my time with others” Cronbach’s α = .59), (2) environmental mastery (e.g., “I am quite good at managing the responsibilities of my daily life” Cronbach’s α = .76), (3) self-acceptance (e.g., “I like most aspects of my personality” Cronbach’s α = .76), (4) autonomy (e.g., “I have confidence in my own opinions, even if they are contrary to the general consensus” Cronbach’s α = .51), (5) personal growth (e.g., “For me, life has been a continuous process of learning, changing, and growth” Cronbach’s α = .66), and (6) purpose in life (“Some people wander aimlessly through life, but I am not one of them” Cronbach’s α = .32). In the current study, we also computed a composite psychological well-being score (i.e., the sum of the 18 items; Cronbach’s α = .85).

The Harmony in Life Scale (Kjell et al., 2013a). This instrument assesses a global sense of harmony in one’s life and consists of 5 statements (e.g., “Most aspects of my life are in balance”) for which participants are asked to indicate degree of agreement on a 7-point Likert scale (1 = strongly disagree, 7 = strongly agree). The harmony score was established by summarizing the 5 statements for each participant. Cronbach’s α were .91 in the present study.

Statistical treatment

The procedure to create the affective profiles was originally developed by Archer and colleagues (e.g., see Norlander, Bood & Archer, 2002) by dividing self-reported positive affect and negative affect scores into high and low. In the present study, we used the following cut-off points reported by Schutz and colleagues (Schutz et al., 2013) who used a large population of US-residents: low positive affect = 3.0 or less; high positive affect = 3.1 or above; low negative affect = 1.8 or less; and high negative affect = 1.9 or above.

In the present study the distribution of affective profiles was as follows: 160 self-fulfilling (61 males, 99 females), 66 low affective (23 males, 43 females), 137 high affective (56 males, 81 females), and 137 self-destructive (47 males, 90 females). The first analysis, using SPSS (version 21), was a Multivariate Analysis of Variance (MANOVA) in which the affective profiles and gender were the independent factors and the dependent factors were the six dimensions of psychological well-being, its composite score, and the harmony in life score. To investigate which dimensions of psychological well-being are related to harmony among profiles we performed a path analysis, using AMOS (version 20), in order to estimate interaction/moderation effects between affective profiles as moderator and psychological well-being dimensions as dependent variables upon harmony. The structural equation model of multi-group analysis showed a Chi-square value  = .00; DF  = 00; comparative fit index  = 1.00; incremental fit index  = 1.00 and normed fit index = 1.00.

Results

Differences in psychological well-being and harmony between affective profiles

The affective profiles had a significant effect on the six dimensions of psychological well-being, its composite score, and the harmony score (F (21,1396.08) = 17.75, p < .001, Wilks’ Lambda  = .51, Observed Power = 1.00). The effect of gender (p = .21) and the interaction of affective profiles and gender (p = .13) were not significant. Self-fulfilling individuals scored higher in all psychological well-being dimensions as compared to all the other profiles: positive relations (F(3,492) = 55.31, p < .001, Observed Power = 1.00), environmental mastery (F(3,492) = 91.50, p < .001, Observed Power = 1.00), self-acceptance (F(3,492) = 88.88, p < .001, Observed Power = 1.00), autonomy (F(3,492) = 11.47, p < .001, Observed Power = 1.00), personal growth (F(3,492) = 40.72, p < .001, Observed Power = 1.00), purpose in life (F(3,492) = 17.45, p < .001, Observed Power = 1.00). The only exception was autonomy, in which no difference was found between the low affective and self-fulfilling groups; and for purpose in life, in which no difference was found between high affective and self-fulfilling groups (see Table 2). Instead, low affective scored higher in autonomy compared to self-destructive individuals, while high affective scored higher in purpose in life compared to both low affective and self-destructive individuals.

Table 2 Mean scores and sd in all six psychological well-being dimensions, psychological well-being total score, and harmony in life score for each affective profile.

	Self-destructive n= 137	Low affective n= 66	High affective n= 137	Self-fulfilling n= 160	
Positive relations with others	3.57 ± .99	4.06 ± .90a	4.25 ± .96a	4.97 ± .91a,b,c	
Environmental mastery	3.11 ± 1.02	4.16 ± 1.08a	4.10 ± .88a	4.92 ± .72a,b,c	
Self-acceptance	2.88 ± 1.06	3.88 ± 1.08a	4.00 ± .98a	4.80 ± .85a,b,c	
Autonomy	4.22 ± .92	4.60 ± .88a	4.41 ± .85	4.81 ± .80a,c	
Personal growth	4.32 ± .97	4.44 ± 1.03	4.96 ± .79a,b	5.31 ± .64a,b,c	
Purpose in life	4.06 ± .93	4.06 ± .81	4.45 ± .84a,b	4.68 ± .78a,b	
Composite psychological well-being	3.69 ± .61	4.20 ± .66a	4.37 ± .54a	4.91 ± .48a,b,c	
Harmony in life	3.25 ± 1.28	4.40 ± 1.27a	4.67 ± 1.28a	4.62 ± .87a,b,c	
Notes.

Values represent mean scores ± SD. P < .01.

a Bonferroni test: higher compared to the self-destructive.

b Bonferroni test: higher compared to the low affective.

c Bonferroni test: higher compared to the high affective.

Nevertheless, regarding the psychological well-being composite score, self-fulfilling individuals scored higher than all the other three affective profiles (F(3,492) = 113.53, p < .001, Observed Power = 1.00), while both the low and high affective individuals scored higher than the self-destructive individuals (see Table 2). With regard to the harmony in life score, self-fulfilling individuals scored higher than all the other three affective profiles (F(3,492) = 93.06, p < .001, Observed Power = 1.00). As for the psychological well-being composite score, the low and high affective individuals reported higher harmony score than the self-destructive individuals (see Table 2).

Multi-group moderation analysis

Four multi-group moderation analyses with the 6 dimensions of psychological well-being as predictors and the harmony in life as the dependent variable showed that 47% to 66% of the variance of the harmony in life is explained by the psychological well-being via the four different affective profiles (see Table 3). Harmony in life was significantly predicted by environmental mastery and self-acceptance across all affective profiles (see Figs. 1–4). However, for the low affective group high purpose in life predicted low levels of harmony in life (see Fig. 2).

Figure 1 Structural equation model of the six dimensions of psychological well-being and harmony in life via the self-destructive group.

All correlations (between different psychological well-being dimensions) and all paths (from the six dimensions of psychological well-being to harmony in life) and their standardized parameter estimates. Chi-square  = .00; DF  = 00; comparative fit index = 1.00; incremental fit index = 1.00 and normed fit index = 1.00. e = error. Red standardized parameter estimates of regression weights are significant at the p <.001 level (n = 137).

Table 3 Structural coefficients for the structural equation model of multi-group moderation between affective profiles as moderator and psychological well-being dimensions on harmony in life.

Self-destructive n = 137	
Predictor	Outcome	β	SE	B	P	
Positive relations	Harmony in life	.12	.08	.10	.12	
Environmental mastery	.46	.10	.37	<.001	
Self-acceptance	.50	.09	.42	<.001	
Autonomy	.00	.08	.00	.95	
Personal growth	.09	.09	.07	.30	
Purpose in life	-.06	.09	-.04	.49	
R 2	.55					
Low affective n =66	
Predictor	Outcome	β	SE	B	P	
Positive relations	Harmony in life	-.17	.12	-.12	.15	
Environmental mastery	.49	.12	.42	<.001	
Self-acceptance	.67	.12	.57	<.001	
Autonomy	.14	.12	.09	.24	
Personal growth	.14	.11	.12	.19	
Purpose in life	-.58	.15	-.37	<.001	
R 2	.66					
High affective n =137	
Predictor	Outcome	β	SE	B	P	
Positive relations	Harmony in life	.18	.10	.13	.07	
Environmental mastery	.79	.11	.54	<.001	
Self-acceptance	.23	.11	.17	<.05	
Autonomy	-.13	.10	-.08	.19	
Personal growth	.14	.11	.09	.22	
Purpose in life	-.21	.10	-.14	.05	
R 2	.47					
Self-fulfilling n =160	
Predictor	Outcome	β	SE	B	P	
Positive relations	Harmony in life	-.05	.06	-.06	.39	
Environmental mastery	.48	.10	.40	<.001	
Self-acceptance	.41	.08	.40	<.001	
Autonomy	-.07	.07	-.06	.31	
Personal growth	.05	.09	.03	.59	
Purpose in life	-.01	.07	-.01	.93	
R 2	.47					
Notes.

Significant regression weight are shown in bold type.

Discussion

The aim of the present study was twofold: (1) to investigate differences between affective profiles in psychological well-being dimensions and harmony and (2) to investigate how psychological well-being dimensions relate to harmony within the four affective profiles. Overall the results revealed that affective profiles systematically relate to the psychological well-being dimensions as well as harmony. Individuals in the self-destructive group reported the lowest levels of psychological well-being and harmony in life; meanwhile individuals classified as self-fulfilling reported higher levels of psychological well-being and harmony in life, compared to all the other three affective profiles. The results are summarized in Fig. 5.

Figure 2 Structural equation model of the six dimensions of psychological well-being and harmony in life via low affective group.

All correlations (between different psychological well-being dimensions) and all paths (from the six dimensions of psychological well-being to harmony in life) and their standardized parameter estimates. Chi-square = .00; DF = 00; comparative fit index = 1.00; incremental fit index = 1.00 and normed fit index = 1.00. e = error. Red standardized parameter estimates of regression weights are significant at the p < .001 level (n = 66).

Figure 3 Structural equation model of the six dimensions of psychological well-being and harmony in life via high affective group.

All correlations (between different psychological well-being dimensions) and all paths (from the six dimensions of psychological well-being to harmony in life) and their standardized parameter estimates. Chi-square = .00; DF = 00; comparative fit index = 1.00; incremental fit index = 1.00 and normed fit index = 1.00. e = error. Red standardized parameter estimates of regression weights are significant at the p < .001 level and blue standardized parameter estimates of regression weights are significant at the p < .05 level (n = 137).

Figure 4 Structural equation model of the six dimensions of psychological well-being and harmony in life via self-fulfilling group.

All correlations (between different psychological well-being dimensions) and all paths (from the six dimensions of psychological well-being to harmony in life) and their standardized parameter estimates. Chi-square = .00; DF = 00; comparative fit index = 1.00; incremental fit index = 1.00 and normed fit index = 1.00. e = error. Red standardized parameter estimates of regression weights are significant at the p < .001 level (n = 160).

Nevertheless, two exceptions to this rule were found for the autonomy and the personal growth dimensions of psychological well-being. First of all, low affective individuals reported higher autonomy than those categorized as self-destructive and not significantly lower than self-fulfilling individuals. In other words, despite experiencing low positive affect, low affective individuals seem to feel confident about their own opinions even if those opinions are in contrast to the general consensus. Garcia and colleagues (Garcia et al., 2010) have earlier suggested that the low affective profiles “go their own way” by using different emotion regulation strategies to avoid displeasure, which also serve as a strategy to feel pleasure (see Higgins, 1997). These researchers showed that low affective individuals actually attenuated their reaction to both negative and positive stimuli by engaging their attention to neutral stimuli (Garcia et al., 2010). This might also explain how individuals categorized as low affective avoid stress in their life (Norlander, Johansson & Bood, 2005), which is in line with the second exception regarding the personal growth dimension and high affective individuals.

Figure 5 Summary of the results showing the differences between affective profiles in the 6 dimensions of psychological well-being and harmony in life.

High affective individuals reported significantly higher levels of personal growth than both self-destructives and low affectives, but still significantly lower than self-fulfilling individuals. Indeed, seeing life as an opportunity to learn and continually grow throughout life is at times rewarding involving positive emotions; but at other times it can be challenging and potentially stressful involving negative emotions (Ryff & Singer, 1998). For instance, compared to low affective, high affective individuals cope better with induced stress (Norlander, Bood & Archer, 2002) but at the same time report more stress in their life compared to both low affective and self-fulfilling individuals (Norlander, Johansson & Bood, 2005). Although autonomy can be considered as an agentic dimension in which low affectives scored higher than self-destructive, low affectives seem to instead show difficulties with stress that is induced by their surroundings (Norlander, Bood & Archer, 2002), not by peer pressure when they choose to “go their own way” and exert their autonomy (Garcia et al., 2010). In other words, experiencing life as a growing experience and a greater purpose, which is related to high affectivity, might be one of the experiences low affective individuals will try to avoid.

Across affective profiles, harmony in life is related to both self-acceptance and environmental mastery. This suggests that a harmonious life might come from accepting all parts of the self and one’s past as well as from the individual’s ability to fit in with environments suitable for their strengths. This is actually a good description of the process of empowerment, a process in which the individual is strengthened to be proactive, non-judgemental, responsible for his own actions, in control of her/his own life, and responsible by helping others (Jimmefors, Garcia & Archer, in press). It is worth pointing out that as a concept, harmony stresses accepting and adapting to the surroundings while environmental mastery emphasizes creating and choosing surroundings. Generally though, these two dimensions are seen to define adaptation to the self and to the environment, which in turn has been associated to the individual’s level of self-awareness (Cloninger, 2004). High levels on the agentic dimension purpose in life were, however, related to low levels of harmony in life among low affective individuals.

The purpose in life dimension comprises attitudes such as goal-setting and planning one’s future. This approach to life can be seen as striving to promote pleasure and avoiding displeasure by accomplishments—the individual focuses on reaching goals for the anticipated rewarding experience of achieving them but also because of the expected pain of failure (Higgins, 1997). This approaching focused behaviour stands in contrast to low affectives’ avoidance tendency (Garcia et al., 2010). Indeed, individuals feel more at ease and in balance when using strategies that are attuned with their approaching/avoiding tendencies (Higgins, 1997). Indeed, low affective individuals scored lower than high positive affect individuals (i.e., high affective and self-fulfilling) in the purpose in life dimension. We suggest that this does not mean that low affectives should avoid having a purpose in life. After all, the absence of positive emotions, for example, is more predictive of subsequent mortality and morbidity than the presence of negative emotions (Cloninger, 2004; Cloninger, 2006; Huppert & Whittington, 2003). At times individuals might need to loosen the status quo in order to promote positive emotions and resilience, while at times individuals need to focus on being in harmony with their environment.

Limitations and inquiries for further research

Although different studies suggest that the quality of the data collected through MTurk meets academic standards and is demographically diverse (Buhrmester, Kwang & Gosling, 2011; Paolacci, Chandler & Ipeirotis, 2010; Horton, Rand & Zeckhauser, 2011) it is plausible to point out some potential issues, such as workers’ attention levels, cross-talk between participants, and the fact that participants get remuneration for their answers (Buhrmester, Kwang & Gosling, 2011). Nevertheless, MTurk is not only suggested as a valid tool for collecting data using personality scales (Buhrmester, Kwang & Gosling, 2011), but also health measures using MTurk data shows satisfactory internal as well as test-retest reliability (Shapiro, Chandler & Mueller, 2013), and payment amount does not seem to affect data quality (Buhrmester, Kwang & Gosling, 2011). Moreover, remuneration in MTurk is usually small and workers report being intrinsically motivated (e.g., for enjoyment) to take part in surveys (Buhrmester, Kwang & Gosling, 2011).

With regard to the psychological well-being dimensions, some showed low reliability (e.g., purpose in life Cronbach’s α = .32); which potentially may question some of the findings. Nevertheless, the internal reliability of the short version used in the present study is almost the same as those obtained by Clarke and colleagues (Clarke et al., 2001). For instance, descriptive data generated with this short measure are consistent with those found with the larger, more reliable 120-item version (Clarke et al., 2001). Moreover, future research needs to randomize the order in which the instruments are presented to participants to ensure that responses to survey questions are not affected by the order of the instruments (Lavrakas, 2008).

Nevertheless, one remaining question is why the positive relations with others dimension was not associated to the sense of a harmonious life. After all, positive and warmth relations with significant others are constantly associated with a happy and satisfied life (e.g., Garcia & Sikström, 2013). In addition -after self-acceptance and environmental mastery- positive relations with others has been found to show the third strongest correlation to harmony (Kjell et al., 2013a). However, it has previously been argued that the (Kjell et al., 2013a). However, it has previously been argued that the positive relations with others dimension involves a rather self-centered perspective to relationships (e.g., Christopher, 1999), which perhaps makes it less in tune with the concept of harmony in life. Another explanation might be that creating and keeping social relationships involves both tolerance and empathy towards others (Cloninger, 2004). While social tolerance might involve adaptation to one’s environment, empathy involves putting oneself in the place of others and perhaps disturbing one’s inner harmony.

In addition, it has been argued that affect as measured in this study involves rather self-centered and high arousal emotions (e.g., Christopher, 1999; Russell & Feldman Barrett, 1999; Yik, Russell & Feldman Barrett, 1999); whilst leaving out more other-centered and low arousal emotions such as feeling compassion, at peace, and love, which might be more in tune with being in harmony. Future research might benefit from also employing an emotion instrument more in line with harmony as this might potentially enrich the investigations of the affective profiles.

Finally, it is plausible to criticize the validity of the procedure used to differentiate the four affective profiles scores just-above and just-below the median become high and low by fiat, not by reality (Schutz et al., 2013). Nevertheless, MacDonald & Kormi-Nouri (2013) used k-means cluster analysis to test if the affective profiles model emerged as theorized by Archer and colleagues. The affective profiles model was replicated using the k-means cluster analysis and the four affective profiles emerged as the combinations of high vs. low affectivity. The procedure used by these researchers is useful for person-oriented analyses (see Bergman, Magnusson & El-Khouri, 2003), thus, suggesting the original procedure by Archer is valid.

Final remarks

The self-fulfilling state, defined as frequently experiencing positive emotions and infrequently experiencing negative emotions, is not only related to more life satisfaction but also to the sense of a harmonious life. Further, an approach focus in life seems to relate to less harmony in life for individuals who prefer to avoid displeasure and staying in a low affective state. Importantly, acceptance of the self as well as environmental mastery might enable individuals with different affective profiles to have the sense of harmony in life.

“He who lives in harmony with himself, lives in harmony with the universe”

Marcus Aurelius

Additional Information and Declarations

Competing Interests

Author Contributions

The authors declare there were no competing interests.

Danilo Garcia conceived and designed the experiments, performed the experiments, analyzed the data, contributed reagents/materials/analysis tools, wrote the paper.

Ali Al Nima performed the experiments, analyzed the data, contributed reagents/materials/analysis tools, wrote the paper.

Oscar N.E. Kjell wrote the paper.

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
