# Peer review of "The affective profiles, psychological well-being, and harmony: environmental mastery and self-acceptance predict the sense of a harmonious life"

_PeerJ, doi:10.7717/peerj.259_

## Round 0.1 · original submission · Major Revisions

Dear Authors, The 4 peer reviewers have given their suggestions which require major rewriting both in the material and methods as well as in the discussion sections.

Reviewer 1 ·

Basic reporting

- parts of manuscript can be rewritten to allow more clear text for readers (e.g. the first and second paragraphs of "Limitations and inquiries for further research").
- the article misses to elaborate on definition of happiness and well-being as it is based on the differences between happiness and well-being.

Experimental design

No Comments.

Validity of the findings

- As mentioned by authors, some dimensions of Psychological Well-being shows low reliability such as "Purpose in life" with only Cronbach’s α = .32, which may question some of the findings.
- Although authors touched upon the usage of their online data collection system and its shortcomings, it seems that they need to elaborate more on how participation to earn money can be considered as voluntary and how their data will not be biased because of possibly specific demography of people who choose to register in such websites.
- I could not find out how the authors have concluded their article by this sentence: "acceptance of the self as well as environmental mastery might enable the individual to move from a self-destructive to a self-fulfilling state that involves the sense of harmony in life". As there is no evidence in their findings regarding the journey from being a self-self-destrcuive to become a self-fulfilling person.

Additional comments

All in all, I feel like your article was worth-reading and the novelty of some of the instruments used in this article (harmony in life and affective profiles) and statistical methods used to handle the research questions was quite notable. However, broader literature review, using other authors viewpoints , and comparing their finding with yours will definitely add some depth to your introduction, findings, and discussions.

Reviewer 2 ·

Basic reporting

Correct text errors such as problems with dots over “o” and “a” in line 66; 109 change to either “life satisfaction” or “satisfaction with life” line 109.
Go over the list of reference (for example: lines 346, 356, 364, 365, 380, 405, 411, 435, 440, 450 462, v469, 475, 496, 509).

Experimental design

No comments

Validity of the findings

No comments

Reviewer 3 ·

Basic reporting

No Comments

Experimental design

The authers are using three self-report measures, but the authers do not present in which order the measures are distributed to the participants. To avoid bias in the order in which the measures are answered, the authers should randomly assign these measures to the participants.

Validity of the findings

No Comments

Additional comments

This is a well written article, where previous research are presented and well interweaved with the present study. The results and discussion are presented in a clear manner. The authers have good insight to the study´s limitations and present valid suggestions for future research.

·

Basic reporting

no comments

Experimental design

no comments

Validity of the findings

no comments

Additional comments

I think the authors here have an excellent opportunity here to discuss the advantages of harmony in endowering empowerment to individuals, particularly since other attributes measured here suggest the confluence of empowering entities.

---

## Round 0.2 · accepted · Accept

Thank you for the revisions made to the manuscript that has lead to it's acceptance

Reviewer 1 ·

Basic reporting

No Comments.

Experimental design

No Comments.

Validity of the findings

No Comments.

Additional comments

I think you have covered the most important points of my previous revision. Good job!

·

Basic reporting

no comment

Experimental design

no comment

Validity of the findings

no comment

Additional comments

good paper